# Monoclonal antibodies targeting sites in respiratory syncytial virus attachment G protein provide protection against RSV-A and RSV-B in mice

Youri Lee [1], Laura Klenow[1], Elizabeth M. Coyle[1], Gabrielle Grubbs [1], Hana Golding[1] & Surender Khurana [1] ✉

Currently, only Palivizumab and Nirsevimab that target the respiratory syncytical virus (RSV) fusion protein are licensed for pre-treatment of infants. Glycoprotein-targeting antibodies may also provide protection against RSV. In this study, we generate monoclonal antibodies from mice immunized with G proteins from RSV-A2 and RSV-B1 strains. These monoclonal antibodies recognize six unique antigenic classes (G0-G5). None of the anti-G monoclonal antibodies neutralize RSV-A2 or RSV-B1 in vitro. In mice challenged with either RSV-A2 line 19 F or RSV-B1, one day after treatment with anti-G monoclonal antibodies, all monoclonal antibodies reduce lung pathology and significantly reduce lung infectious viral titers by more than 2 logs on day 5 post-RSV challenge. RSV dissemination in the lungs was variable and correlated with lung pathology. We demonstrate new cross-protective anti-G monoclonal antibodies targeting multiple sites including conformation-dependent class G0 MAb 77D2, CCD-specific class G1 MAb 40D8, and carboxy terminus of CCD class G5 MAb 7H11, to support development of G-targeting monoclonal antibodies against RSV.

Respiratory syncytial virus (RSV) is the major cause of lower respiratory tract disease in infants and young children[1,2], resulting in approximately 3.2 million hospitalizations and 118,200 deaths per year worldwide in children under the age of 5 years[2]. RSV has been classified into two antigenically distinct subtypes RSV A and RSV B, with these strains not only having antigenic differences, but differing clinical characteristics as well[3,4]. RSV subtypes often co-circulate during the same season and have equivalent severity[5].

RSV contains two major surface glycoproteins, the attachment (G) and fusion (F) glycoproteins, which are both targets of neutralizing and/or protective antibodies[6–8]. RSV F is highly conserved between RSV A and B subtypes, and a recently approved vaccine against RSV in older adults demonstrated cross-subtype protection after vaccination with adjuvanted pre-fusion stabilized F protein from RSV A2[9]. There are two licensed monoclonal antibodies (MAbs), Palivizumab and Nirsevimab, both target the fusion (F) protein, which can reduce disease in high-risk premature-birth infants or healthy infants 8 months of age and younger, respectively, if administered prior to RSV infection[10]. Several other MAbs targeting the F protein are being evaluated for the prevention of RSV in infants and children[11].

RSV G protein is more variable. In addition to its function as an attachment protein, RSV G is a potential contributor to immune modulation and disease pathogenesis[12,13]. Most of the vaccine and therapeutics targeting RSV G are focused on the central conserved domain (CCD) of G and the adjacent fractalkine-like CX3C motif[14,15]. Antibodies against the CX3C motif were suggested to provide

[1]Division of Viral Products, Center for Biologics Evaluation and Research (CBER), FDA, Silver Spring, MD 20993, USA.
✉e-mail: Surender.Khurana@fda.hhs.gov

protection against the RSV inflammatory disease[16–18]. Anti-G MAb 131-2 G targeting CCD motif was shown to block the interaction of RSV G protein with surface CX3CR1 and to block RSV G protein induced chemotaxis[12]. In vivo, it protected animals from RSV disease and lung pathology[17,19,20]. However, MAb 131-2G and other MAbs targeting the CCD region do not neutralize RSV in in vitro neutralization assays[21].

In a previous study, we elucidated the complete antibody epitope repertoire following primary RSV infection in infants using RSV genome fragment phage display libraries (GFPDL) in different age groups. That study demonstrated an increase of G specific binding antibodies over time[22]. In RSV-G, the bound phages displayed epitopes spanning the entire ectodomain of RSV-G in addition to the conserved central domain (CCD; aa residues 172-186). To better understand the potential role of antibodies targeting various antigenic sites across RSV-G, in addition to the CCD motif, we generated a panel of murine MAbs against the G protein of RSV A and RSV B. These MAbs were evaluated in neutralization assays, strain-specificity, and epitope mapping using ELISA and SPR technologies with RSV G peptides and protein domains and were evaluated for their protective efficacy in in vivo challenge mouse model against RSV-A2 and RSV-B1.

## Results

### Generation and identification of six classes of RSV G-specific monoclonal antibodies

Six-week-old female C57BL/6 mice were immunized intramuscularly twice with recombinant non-glycosylated G proteins produced in *E. coli*, from either RSV-A2 strain termed REG-A ($n = 6$), or RSV-B1 strain termed REG-B ($n = 6$)[23], at 28-days interval. Mouse spleens were isolated at 7-days following the second vaccination and used to generate hybridomas, followed by single cell cloning. Clones were screened against glycosylated forms of RSV-G protein produced in mammalian cells from either RSV-A2 (RMG-A2) or RSV-B1 (RMG-B1)[23,24] by ELISA. The clones that showed strong anti-RSV G antibody binding to any of the RSV G proteins were further expanded into large flasks and used for antibody purification using Protein A chromatography.

All MAbs were subjected to a multi-tier epitope mapping and specificity analysis using ELISA or SPR technologies as summarized in Table 1. Cross-reactivity of MAbs against G proteins of RSV A2 and B1 strains were determined by ELISA with recombinant glycosylated forms of G protein produced in mammalian cells of RSV-A2 (RMG-A2) or RSV-B1 (RMG-B1) (Supplementary Fig. S1). Fine epitope mapping was performed using G-derived peptides from RSV-A2 strain previously identified using GFPDL analysis of post-RSV infection infant sera (Supplementary Fig. S1)[22].

Six classes (G0-G5) of MAbs were identified (Table 1). Class G0 included conformational-dependent antibodies that bound to the intact glycosylated G proteins but not to any of the individual RSV A2 derived G peptides (SPR binding <10 RU). MAb 12F12 bound RMG-A2 only, while MAbs 68C7, 69C1, and 75F10 bound RMG-B1 only in ELISA. We also measured binding to a CCD-deleted non-glycosylated REG-A2 protein (REG-A2 delCCD) using SPR. MAb 12F12 bound REG-A2 delCCD, while the three RSV-B1 specific G0 MAbs did not bind REG-A2 delCCD. They may target sites that are less conserved between A2 and B1 G proteins (Supplementary Fig. S1). One G0 Mab (77D2) showed strong cross-reactivity against both RMG-A2 and RMG-B1 proteins in ELISA. While both 12F12 and 77D2 showed similar reactivity to RMG-A2 in ELISA, in SPR, 12F12 showed much higher binding to CCD deleted unglycosylated REG-A2 protein (REG-A2 delCCD) than 77D2, suggesting differences in their epitope footprints.

Class G1 MAbs specifically targeted the CCD region (aa 172-186), similar to the previously described MAb 131-2 G (Table 1). All the G1 MAbs demonstrated stronger binding to RMG-A2 than RMG-B1 protein in ELISA and no binding to the CCD-deleted RSV G protein in SPR. The G1 MAbs reacted to CCD peptide (aa residues 172–186) in SPR. All G1

MAbs demonstrated cross-reactivity between RSV-A2 and RSV-B1 G proteins, especially MAb 40D8 and MAb 7H9.

Class G2 included two MAbs (7C6 and 7G6) that reacted with RMG-A2, but not to RMG-B1 in ELISA. These class G2 MAbs reacted strongly to the CCD-deleted REG-A2 protein in SPR and bind primarily to N-terminal peptide (aa residues 61–90) of RSV-G.

Class G3 MAb 48E2 targets a discontinuous epitope consisting of two peptides (residues 129–152 and 169–207) that flank the CCD motif and form the stem of the CCD loop. This MAb is cross-reactive against both RMG-A2 and RMG-B1 in ELISA and binds to CCD-deleted REG-A2 protein in SPR (Table 1).

Class G4 MAb 72E6 binds much stronger to RMG-B1 than to RMG-A2. It also targets a discontinuous epitope flanking the CCD motif consisting of the peptides upstream and downstream of the CCD loop, but less strong than the cross-reactive MAb 48E2 (Table 1). This weak binding may reflect amino acid differences between A2 and B1 in these regions (Supplementary Fig. S1).

Class G5 MAbs 7H11 and 23B4 bind strongly to RMG-A2 and to a lesser degree with RMG-B1. These G5 MAbs bind CCD-deleted REG-A2 protein (REG-A2 delCCD) and to the peptide encompassing residues 169-297 (downstream of CCD) in SPR (Table 1).

These data suggested that vaccination of mice with non-glycosylated G proteins from RSV-A2 and RSV-B1 elicited MAbs that bind strongly to the glycosylated G proteins derived from the RSV A2 and RSV B1 strains. Further mapping using REG-A2 delCCD protein and peptides spanning the RSV A2 G protein in SPR, identified six classes of antibodies. In addition to targeting the CCD or epitopes upstream or downstream of the CCD (at the stem of CCD loop), we identified Class G0 MAbs that bound only glycosylated intact RSV G proteins from either subtype that did not bind linear RSV-G peptides, and an antibody binding to a site in the N-terminal region. Similar to MAb 131-2 G, all the isolated MAbs did not neutralize RSV-A2 or RSV-B1 in vitro (Supplementary Table S1).

### Protective efficacy of MAbs against RSV A2 and RSV B1 in mice challenge model: impact of prophylactic MAbs treatment

To determine the prophylactic protective efficacy of the MAbs, 4–6-week-old female BALB/c mice (5 mice per group) were intraperitoneally (i.p.) injected with 20 µg/mouse of RSV G specific MAbs, or MAb 131-2 G, or with PBS (negative control) (Fig. 1). The MAbs that were used for pre-treatment prior to RSV-A2 and RSV-B1 challenge were selected based on their epitope mapping, representing classes G0-G5 (Table 1). The protective efficacy of these MAbs, which target different sites in RSV-G protein, was determined by challenging mice with RSV-A2 line 19 F expressing firefly luciferase [RSV-A2-L19-FFL] or RSV B1 expressing firefly luciferase (RSV-B1-FFL) that allows to track RSV infection in mice using live imaging, as previously described[25]. One day after MAb administration, mice were intranasally (i.n.) infected with $1 \times 10^6$ PFU of RSV-A2-L19-FFL or RSV-B1-FFL as previously described[24]. RSV dissemination in the nasal cavity and lungs were inferred using fluorescence measurements obtained via whole body live imaging. Mice were sacrificed 5 days post-RSV challenge (the day of peak viral load). RSV infectious viral titers were measured by plaque-forming units (PFU) in the lungs. Additionally, the lungs were used for histopathological evaluations (Fig. 1a, b).

Infectious replicating RSV titers in lungs were determined by immune-plaque assay in Hep-2 cells. Both RSV-A2 and RSV-B1 replicated in lungs with peak titers >$10^4$ PFU/gram tissue on day 5 post-viral challenge. MAb 131-2 G blocked the infectious virus titers in the lungs of animals infected with either RSV-A2 or RSV-B1 (Fig. 2a, b). Surprisingly, all anti-G MAbs significantly reduced infectious viral loads by more than 2 logs on day 5 post-RSV challenge (Fig. 2a, b).

For histopathological analysis, the lung sections on day 5 post-RSV challenge were examined and scored by a certified veterinary pathologist, blinded to the treatment groups, including the following

**Table 1 | Classification of MAb reactivity against G protein of RSV-A2 and RSV-B1 strains**

| MAb classes | MAb ID | RSV neutralization in vitro | ELISA end-point titer* | | Max RU binding to RSV G proteins and peptides in SPR** | | | | | | | | | | |
|---|---|---|---|---|---|---|---|---|---|---|---|---|---|---|---|
| | | | RMG-A2 | RMG-B1 | REG-A2 delCCD | 33-61 | 61-90 | 90-110 | 129-152 | 172-186 | 169-207 | 236-263 | 263-298 | Zika NS5 |
| Class G0 (Conformational intact G) | 68C7 | No | NR | 1,024,000 | 1.4 | 0.0 | 0.0 | 0.0 | 0.0 | 0.0 | 5.2 | 0.0 | 0.0 | 0.0 |
| | 12F12 | No | 512,000 | NR | 2263.7 | 0.0 | 0.0 | 0.0 | 3.3 | 0.0 | 0.3 | 0.0 | 5.8 | 0.0 |
| | 69C1 | No | NR | 1,024,000 | 0.0 | 0.9 | 0.0 | 3.0 | 1.0 | 1.5 | 5.9 | 0.0 | 0.0 | 0.0 |
| | 75F10 | No | NR | 1,024,000 | 0.0 | 0.0 | 0.0 | 1.4 | 0.3 | 0.4 | 3.2 | 0.0 | 0.0 | 0.2 |
| | 77D2 | No | 512,000 | 512,000 | 189.4 | 0.0 | 0.0 | 1.5 | 0.6 | 2.7 | 3.8 | 0.9 | 0.5 | 1.9 |
| Class G1 (CCD motif) | 40D8 | No | 512,000 | 64,000 | 0.5 | 4.1 | 3.5 | 6.2 | 3.0 | 3188.9 | 0.0 | 0.0 | 0.0 | 0.0 |
| | 1D9 | No | 512,000 | 4,000 | 0.0 | 9.5 | 5.2 | 6.0 | 7.5 | 3029.4 | 1.2 | 1.3 | 6.2 | 5.2 |
| | 7H9 | No | 512,000 | 32,000 | 0.0 | 0.3 | 2.8 | 2.6 | 8.3 | 3335.7 | 0.0 | 0.0 | 1.1 | 0.0 |
| | 12G11 | No | 512,000 | 4,000 | 0.0 | 4.9 | 4.1 | 4.3 | 6.8 | 3582.3 | 0.0 | 4.1 | 7.0 | 1.6 |
| | 22B11 | No | 1,024,000 | 8,000 | 0.4 | 4.7 | 4.0 | 4.3 | 3.6 | 3464.8 | 0.0 | 0.0 | 0.0 | 1.0 |
| | 36E10 | No | 512,000 | 8,000 | 0.0 | 1.7 | 3.6 | 2.8 | 7.3 | 2705.1 | 0.0 | 0.0 | 0.0 | 0.0 |
| | 131-2 G | No | 512,000 | 512,000 | 1.5 | 6.0 | 5.9 | 7.1 | 3.3 | 3795.1 | 0.0 | 0.0 | 0.0 | 0.0 |
| Class G2 (N-term) | 7C6 | No | 512,000 | NR | 1250.1 | 74.7 | 2910.6 | 0.0 | 4.5 | 3.1 | 1.8 | 0.0 | 1.1 | 0.0 |
| | 7G6 | No | 512,000 | NR | 1627.4 | 0.0 | 3192.7 | 0.0 | 0.0 | 1.8 | 2.0 | 0.0 | 0.6 | 0.0 |
| Class G3; Dual A2 & B1 binding (Strong N + C-Term CCD) | 48E2 | No | 1,024,000 | 512,000 | 1026.7 | 1.4 | 0.0 | 1.1 | 2226.6 | 4.7 | 2622.9 | 0.0 | 0.0 | 0.0 |
| Class G4 (Weak N + C-Term CCD) | 72E6 | No | 8,000 | 512,000 | 0.0 | 0.0 | 0.0 | 0.0 | 317.1 | 0.0 | 143.6 | 0.0 | 0.0 | 0.2 |
| Class G5 (C-Term CCD) | 7H11 | No | 1,024,000 | 256,000 | 321.2 | 0.0 | 1.5 | 2.0 | 4.3 | 4.5 | 2328.3 | 0.0 | 0.0 | 0.0 |
| | 23B4 | No | 512,000 | 4,000 | 683.6 | 0.0 | 0.0 | 0.0 | 0.0 | 0.0 | 166.6 | 2.4 | 0.0 | 1.4 |

* ELISA end-point titers of the Mabs were determined as the reciprocal of the highest dilution providing an optical density (OD) twice that of the negative control.

** SPR binding to protein/peptides are shown as maximum resonance units (RU).

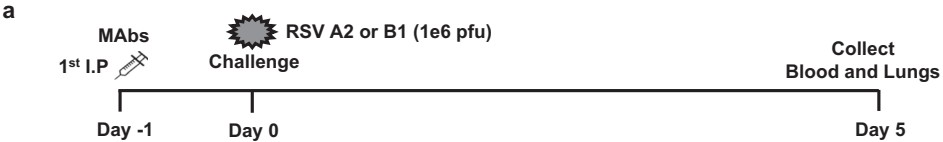

| Class | Group | Dose | Days of imaging | Day of Harvest | Challenge RSV dose |
|---|---|---|---|---|---|
| **RSV-A2 challenge studies** | | | | | |
|  | Uninfected |  | 1-5 dpi | 5 dpi | 1 X 10⁶ |
| G0 (Conformational intact G) | **77D2** | 20 µg | 1-5 dpi | 5 dpi | 1 X 10⁶ |
| G1 (CCD motif) | **40D8** | 20 µg | 1-5 dpi | 5 dpi | 1 X 10⁶ |
|  | 1D9 | 20 µg | 1-5 dpi | 5 dpi | 1 X 10⁶ |
|  | 7H9 | 20 µg | 1-5 dpi | 5 dpi | 1 X 10⁶ |
|  | 12G11 | 20 µg | 1-5 dpi | 5 dpi | 1 X 10⁶ |
|  | 22B11 | 20 µg | 1-5 dpi | 5 dpi | 1 X 10⁶ |
|  | 36E10 | 20 µg | 1-5 dpi | 5 dpi | 1 X 10⁶ |
| G2 (N-term) | 7G6 | 20 µg | 1-5 dpi | 5 dpi | 1 X 10⁶ |
| G3 (Strong N+C-Term CCD) | **48E2** | 20 µg | 1-5 dpi | 5 dpi | 1 X 10⁶ |
| G5 (C-Term CCD) | **7H11** | 20 µg | 1-5 dpi | 5 dpi | 1 X 10⁶ |
|  | 23B4 | 20 µg | 1-5 dpi | 5 dpi | 1 X 10⁶ |
| Control | 131-2G | 20 µg | 1-5 dpi | 5 dpi | 1 X 10⁶ |
| **RSV-B1 challenge studies** | | | | | |
| G0 (Conformational intact G) | 68C7 | 20 µg | 1-5 dpi | 5 dpi | 1 X 10⁶ |
|  | 75F10 | 20 µg | 1-5 dpi | 5 dpi | 1 X 10⁶ |
|  | **77D2** | 20 µg | 1-5 dpi | 5 dpi | 1 X 10⁶ |
| G1 (CCD motif) | **40D8** | 20 µg | 1-5 dpi | 5 dpi | 1 X 10⁶ |
| G3 (Strong N+C-Term CCD) | **48E2** | 20 µg | 1-5 dpi | 5 dpi | 1 X 10⁶ |
| G4 (Weak N+C-Term CCD) | 72E6 | 20 µg | 1-5 dpi | 5 dpi | 1 X 10⁶ |
| G5 (C-Term CCD) | **7H11** | 20 µg | 1-5 dpi | 5 dpi | 1 X 10⁶ |
| Control | 131-2G | 20 µg | 1-5 dpi | 5 dpi | 1 X 10⁶ |

**Fig. 1 | Anti-G MAbs and RSV challenge studies in BALB/c mice. a** Schematic representation of MAb injection and RSV challenge schedule in mice. **b** Female BALB/c mice ($N = 5$ per group; 4–6 weeks old) were prophylactically treated intraperitoneally (IP) with the indicated MAbs at the 20 mcg/mouse dose or with PBS as a control. MAbs labeled in bold were cross-reactive antibodies that were evaluated in challenge studies against both RSV-A2 and RSV-B1. Twenty-four hours after MAb injection, mice were challenged intranasally with $10^6$ PFU of either RSV rA2-Line-19F-FFL or firefly luciferase expressing RSV B1 virus. In vivo imaging of lungs and the nasal cavity was performed daily for 5 days following RSV infection. Mice were sacrificed on day 5 post-challenge, when lungs, and blood were collected.

categories: epithelial alterations in alveolitis, bronchiolitis, perivascular, and interstitial space[26]. Inflammation and focal aggregates of infiltrating cells were examined and measured using a semiquantitative scale (0–3) (0 = absent; normal), 1 (mild inflammation; <20% of lung affected), 2 (moderate inflammation; 20–40% of lung affected), and 3 = severe; 40–60% lung affected) by light microscope (Fig. S2). The lung histopathology scores for the four attributes were then combined to a scale of 0 to 12.

The lung pathology scores on day 5 following RSV-A2 challenge varied for different MAb-treated animals but were significantly lower than the PBS-treated control animals (Fig. 2c–d), which was similar to MAb 131-2 G treated animals, except for G1 MAbs 7H9 and 22B11, and G3 MAb 48E2-treated animals (Fig. 2c). Interestingly, class G0 MAb 77D2, G1 MAb 40D8, 1D9 and 36E10 as well as G2 MAb 7G6 treated animals showed lower pathology than 131-2G treated animals following RSV-A2 challenge, although these differences did not reach statistical significance (Fig. 2c).

All mice that received prophylactic treatment of MAbs prior to RSV-B1 challenge demonstrated reduced lung pathology scores, similar to MAb 131-2 G treated animals compared with lung pathology observed in the PBS control treated animals (Fig. 2d). Lung pathology scores of mice treated with few MAbs including G1-MAb 40D8 showed stronger reduction and were similar to those observed for uninfected control animals following RSV-B1 challenge.

Together, these finding suggested that the majority of MAbs belonging to various classes, targeting different regions of RSV-A2 and RSV-B1 G proteins given one day prior to challenge, reduced lung pathology to different levels following either RSV-A2 or RSV-B1 challenge in the mouse model.

**Anti-G MAbs protect from RSV dissemination in lungs: Live imaging of viral spread in the lungs of RSV-infected animals**
RSV can spread within the host either via infection of target cells by the viral inoculum or the released RSV particles, but also more efficiently

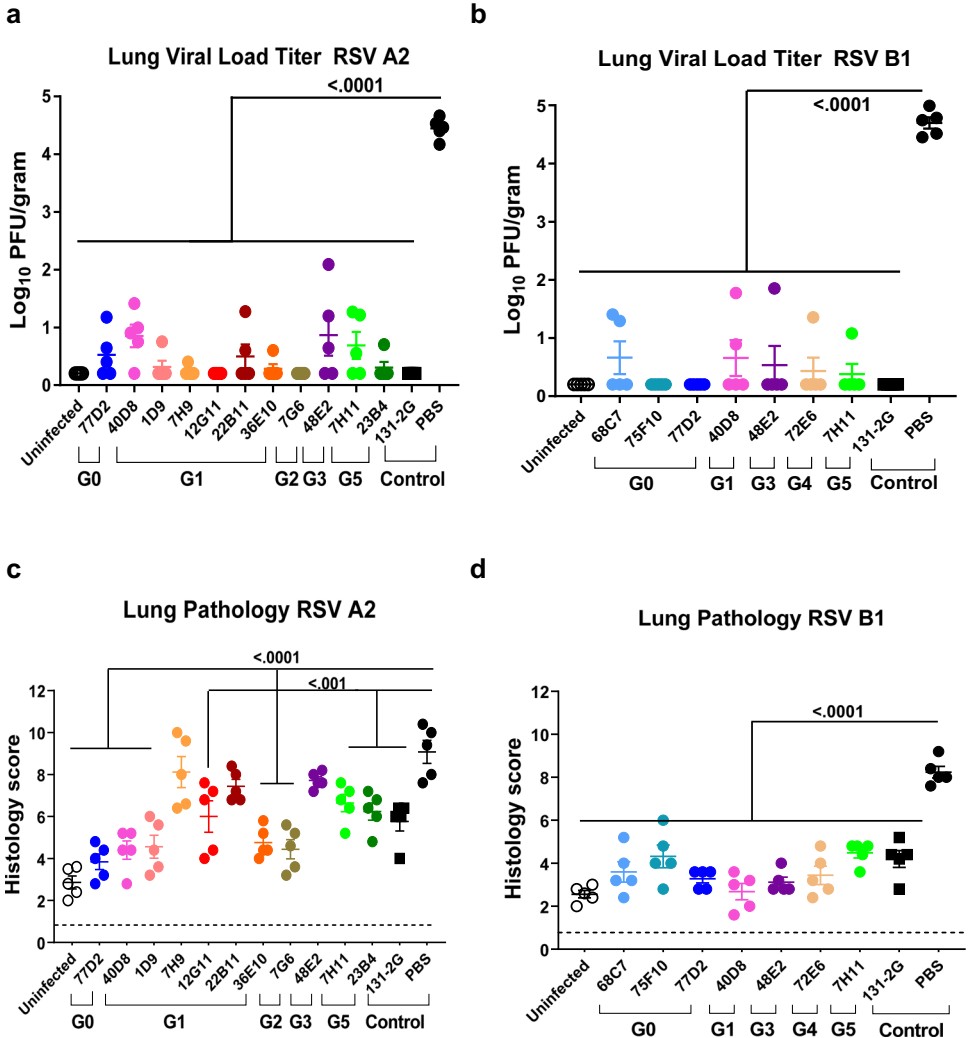

**Fig. 2 | RSV titers and histopathology in the lungs of BALB/c mice at day 5 following RSV challenge.** BALB/c mice (*N* = 5 per group; 4−6 weeks old) were primed i.p. with RSV G-specific MAbs (RSV A2: 36E10, 1D9, 7H9, 22B11, 7G6, 12G11, and 23B4; RSV B1: 68C7, 75F10 and 72E6; or both RSV A2 and RSV B1: 48E2, 7H11, 40D8, 77D2, and 131-2G). Groups of mice treated with various MAbs are shown by different colored symbols. Mice were challenged 24 hours after antibody treatment with RSV-A2-L19-FFL or RSV-B1-FFL i.n., and lung viral titer of RSV-A2-L19-FFL (**a**) and RSV-B1-FFL (**b**) on day 5 post-RSV challenge were determined. (c-d) Lung histopathology on day 5 following RSV-A2 or RSV-B1 virus challenge. Lung tissue of the mice were collected at 5 days after RSV-A2-FFL (**c**) or RSV-B1-FFL (**d**) challenge and

were stained with hematoxylin and eosin. Individual lungs were scored blindly using a 0−3 severity scale for pulmonary inflammation: bronchiolitis (mucous metaplasia of bronchioles), perivasculitis (inflammatory cell infiltration around the small blood vessels), interstitial pneumonia (inflammatory cell infiltration and thickening of alveolar walls), and alveolitis (cells within the alveolar spaces). The scores were subsequently converted to a combined histopathology scale of 0−12. Results are presented as mean ± SEM. Two-way ANOVA with Bonferroni multiple comparisons was performed in GraphPad Prism. The differences were considered statistically significant with a 95% confidence interval when the *p* value was <0.05. The significant *p*-values are shown. Source data are provided with this paper.

via direct cell-to-cell transmission[27]. To understand the dissemination of RSV in untreated and MAb-treated animals, whole-body live imaging of infected mice was performed using IVIS imaging system as previously described[6,25]. None of the MAbs (including MAb 131-2 G) reduced virus transmission in the nasal cavity as measured by Flux (photons/sec) units using live imaging of infected mice (Fig. 3a, b). On day 5, the lung fluxes varied among the MAb-treated animals (Fig. S3). Interestingly, the positive control MAb 131-2G reduced lung fluxes after RSV-B1 (~5-fold) more efficiently than after RSV-A2 infection (~2-fold) and was not statistically different from the PBS (untreated) control animals (Fig. 3c, d). Importantly, in RSV-A2 challenged animals, several MAbs reduced day 5 lung-fluxes more efficiently than 131-2G, including class G0 MAb 77D2, G1 MAbs 40D8, 1D9, 12G11, 22B11, and 36E10, G3 MAb 48E2, as well as G5 MAb 7H11 in RSV-A2 infected animals (Fig. 3c).

For RSV-B1, 4 of the 7 MAbs controlled viral spread in lungs compared with the PBS control (Fig. 3d). Importantly, none of the MAbs-treated animals demonstrated enhanced viral loads compared

with the PBS-treated animals by lung-flux measurements. Interestingly, the cross-reactive antibodies from three different classes: G0 MAb 77D2, G1 MAb 40D8, and class G5 Mab 7H11, demonstrated significant reduction in lung fluxes against both RSV-A2 and RSV-B1.

Correlations were calculated with a Spearman two-tailed test to determine the relationship between lung pathology and RSV spread in lungs (Flux units) or the lung infectious viral titers (PFU). A statistically significant correlation was observed between the lung pathology scores and lung flux measurement on day 5 for individual mice across all groups for RSV-A2 (*p = 0.0317*) (Fig. 4a) and RSV-B1 (*p* = 0.0058; Fig. 4b), but not with lung infectious viral titers (Fig. 4c, d). No significant correlation was observed between lung PFU and lung flux activity at day 5 post-RSV challenge in these mice infected with either RSV-A2 or RSV-B1 (Fig. S4).

Together, our data demonstrates that several anti-G MAbs targeting multiple sites, including conformation-dependent class G0 MAb 77D2, CCD-specific class G1 MAb 40D8, and carboxy terminus of

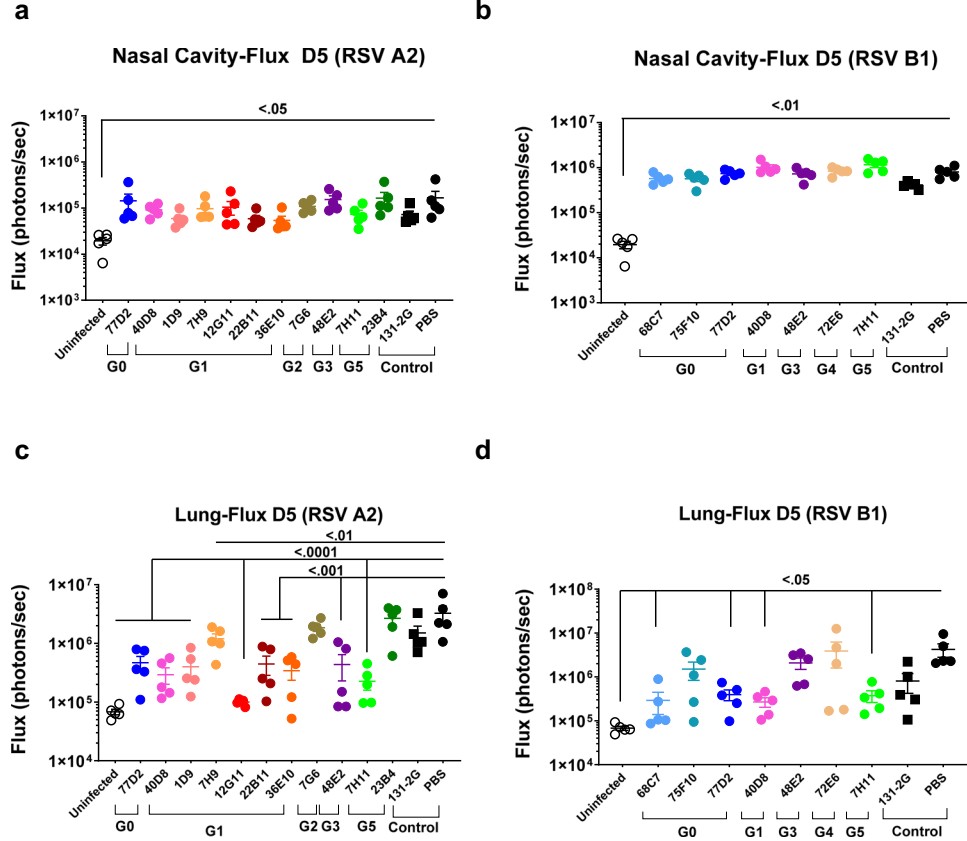

**Fig. 3 | Live imaging of RSV infection in nasal cavity and lung in BALB/c mice at day 5 post-RSV challenge.** MAb treated mice (*N* = 5 per group; 4–6 weeks old) were challenged 24 h afterwards with firefly expressing RSV-A2-L19-FFL or RSV-B1-FFL virus. Groups of mice treated with various MAbs are depicted by different colored symbols. Live whole-body imaging was performed to detect firefly luciferase activity in the nasal cavity (**a**, **b**) and lungs (**c**, **d**) of either RSV-A2-FFL (**a**, **c**) or RSV-

B1-FFL (**b**, **d**) virus expressing firefly luciferase. Graphs represent the quantification of total flux (photons/sec) in these organs on day 5 post-challenge. Results are presented as mean ± SEM. Two-way ANOVA with Bonferroni multiple comparisons was performed in GraphPad Prism. The differences were considered statistically significant with a 95% confidence interval when the *p* value was <0.05. The significant *p*-values are shown.. Source data are provided with this paper.

CCD class G5 MAb 7H11, showed cross-reactive protection from lung pathology and RSV dissemination following challenge with either RSV-A or B subtypes better than MAb 131-2 G (Fig. 5).

## Discussion

RSV is the leading cause of respiratory disease in children worldwide and the primary cause of hospitalization for viral respiratory infections and a major cause of overall mortality in infants and children, especially premature infants[28]. Therefore, effective RSV prevention strategies are needed to address this major public health issue and burden[29]. A recent meta-analysis by Sun et al. concluded that the licensed MAbs, Palivizumab and Nirsevimab, both targeting the F protein, were associated with substantial benefits in the prevention of RSV infection, without a significant increase in adverse events compared with placebo[11].

In a previous study, we found evidence that young infants prior to exposure to RSV have lower titers of anti-G antibodies, compared with anti-F antibodies, reflecting preferential transplacental transfer of anti-F vs. anti-G antibodies. The G-binding antibodies increased 100-fold after primary RSV infection and were mapped to multiple regions in the RSV G protein, in addition to the CCD region[22]. Subsequently, we have shown that un-glycosylated G protein (REG), as well as REG with CCD deletion can elicit protective immunity in mice[23]. Furthermore, several G-derived peptides outside the CCD/CXCR3 induced protective immunity with lower viral loads and pathology scores in RSV challenged animals[6]. These findings suggested that monoclonal antibodies targeting different sites in the G

proteins of RSV type A and type B given prophylactically may provide protection against RSV disease.

In the current study, we generated and evaluated a panel of RSV G-targeting MAbs that were mapped to different sites in RSV G in addition to the CCD motif, for their effectiveness in reducing viral dissemination in the lungs and protection from lung pathology. We used the previously described protective anti-G MAb 131-2 G, which does not neutralize RSV in vitro, as a benchmark[17,19,20]. Epitope mapping and relative prophylactic effectiveness in reducing lung infectious RSV titers, RSV spread in lungs and protection from lung pathology following either RSV-A2 and RSV-B1 infection is schematically summarized in Fig. 5.

Similar to MAb 131-2 G, the new MAbs did not neutralize RSV in vitro. However, in vivo these anti-G MAbs reduced virus dissemination to the lungs either equally or better than 131-2 G following challenge with RSV-A2 or RSV-B1 (Fig. 3). The discrepancy between the in vitro neutralization and the in vivo results for anti-G antibodies is well documented and could be due to the absence of Fc-receptor functions and interactions with effector cells in the in vitro system. The correlation between lung pathology and RSV dissemination (lung flux) but not with infectious viral titer in the lungs is remarkable (Fig. 4) and suggest that viral dissemination may be a more sensitive readout in the mouse model. Following RSV infection, the early inflammatory response is further activated by TLR-2, TLR-3, TLR-4, and TLR-7 followed by the production of cytokines such as interleukin-6 (IL-6), IL-8 and type I and III interferon (IFN) by alveolar macrophages and epithelial cells. This leads to further recruitment of innate immune cells,

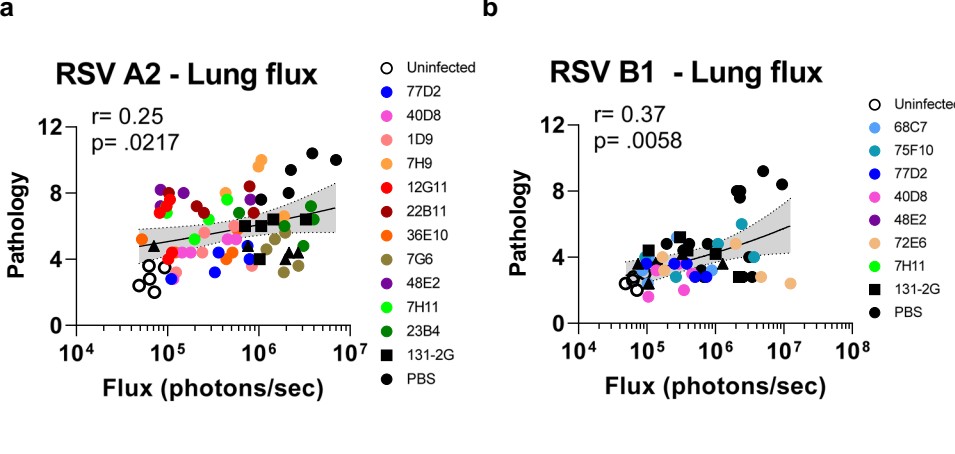

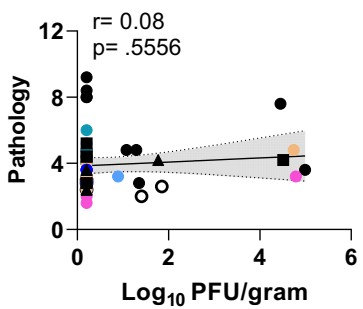

**Fig. 4 | Relationship of lung pathology and lung fluxes or viral load titer on day 5 post-challenge with either RSV-A2 or RSV-B1.** Correlation of lung pathology scores versus bioluminescence flux (**a**, **b**) signal in the infected lungs on day 5 post-challenge or viral load measured by plaque assay in lungs on day 5 post-challenge (**c**, **d**) with either RSV-A2-FFL (**a**, **c**) or RSV-B1-FFL (**b**, **d**) for all MAb treated BALB/c mice (*N* = 5 per group; 4–6 weeks old). Groups of mice treated with various MAbs are denoted by different colored symbols similarly as shown in Figs. 2 and 3. Correlations show Spearman two-tailed test correlation coefficient (*r*) and two-tailed *p* values for all samples. The black line in the scatter plots depict the linear fit with shaded area showing 95% confidence interval. Source data are provided with this paper.

especially neutrophils, monocytes and dendritic cells to the lung. Soluble G protein, NS1and NS2 inhibit the host type I IFN response, indicating that any of these three proteins may be targeted therapeutically[27,30]. Our observations suggests that the lung pathology following RSV infection could be a result of viral dissemination primarily by cell-to-cell spread in the lungs and the strong inflammatory response that follows. RSV spread within the host more efficiently via direct cell-to-cell transmission that may result in the lung pathology observed in animals in the absence of the productive infectious viral particle generated following RSV infection[27]. Several of the new anti-G MAbs were more efficient in restricting the viral dissemination in lungs as measured by live imaging of RSV challenged mice, suggesting a key role for G protein in the cell-to-cell spread of RSV. In the next stage of development, the findings in the mice challenge studies will be expanded to the cotton rat model with contemporary RSV strains from the RSV/A/Ontario and RSV/B/Buenos Aires genotypes to understand the prophylactic and therapeutic application of these MAbs. The protective mechanism of these anti-G MAbs can be further deciphered in other pre-clinical models such as the human airway organoid (lung or nose) that may better recapitulate the human airway.

Importantly, among the more effective MAbs, MAb 77D2 (G0 class), MAb 40D8 (G1 class), and MAb 7H11 (G5 class) are cross-reactive against both RSV-A2 and RSV-B1 (Table 1, Fig. 5). The mechanisms of protection from viral dissemination to the lungs by the new anti-G MAbs, similar to MAb 131-2 G, are not well understood, but may involve

Fc mediated functions including antibody-dependent cellular cytotoxicity (ADCC), antibody-dependent cellular phagocytosis (ADCP), and antibody-dependent complement deposition (ADCD), similar to some anti-F antibodies elicited by infection or vaccination[31]. Similar mechanisms may occur for anti-F MAb treatment and therefore, our study suggests a potential role for combining anti-G and anti-F MAbs as more effective prophylaxis and/or treatment against RSV.

Our findings suggest that in addition to the currently approved F-targeting MAbs, anti-G cross-reactive MAbs can be used as prophylactic MAbs for prevention of RSV disease.

## Methods

### Cell culture and virus production

A549 cells (Cat. No. #CCL-185) were obtained from the American Type Culture Collection (ATCC, Manassas, VA, USA). RSV rA2-Line19F-Firefly Luciferase virus (rRSV-A2-L19-FFL) or RSV-B1-FFL virus expressing the firefly luciferase gene upstream of the NS1 gene were prepared by infecting A549 cell at 0.02 multiplicity of infection (MOI). The virus was harvested by freeze/thaw cycles and were then purified via a sucrose gradient prior to being aliquoted and stored at −80 °C. To determine the titer of the virus stock, immune plaque assays were performed on A549 cells. The optimal challenge dose (10^6 PFU intranasally) and peak days of viral infection was determined by BALB/c model in which viral loads were measured by traditional plaque assay in Hep-2 cells, and by live imaging flux[25].

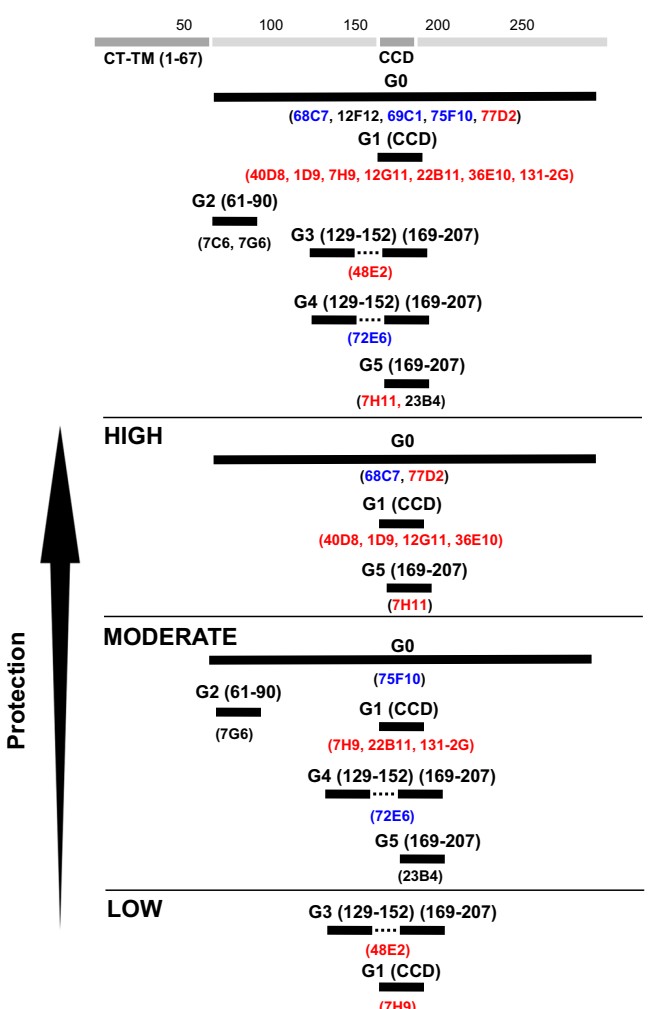

**Fig. 5 | Cross-reactivity and protective efficacy of different classes of anti-G MAbs against RSV-A and RSV-B.** Schematic summarizing the classification of RSV-G binding MAbs by mapping studies and protective efficacy based on lung flux and lung pathology in BALB/c mice challenge studies against RSV-A2 and RSV-B1. All MAbs reduced lung infectious viral titers. High protective MAbs were defined as the MAbs that significantly reduced lung fluxes and lung pathology; moderately protective MAbs were defined as those who reduced lung flux and/or lung pathology (similar to previously described moderately protective MAb 131-3 G), but less efficiently than the highly protective MAbs category; while low protective MAbs only reduced lung infectious viral titers but did not reduce lung flux nor lung pathology.

### Production of recombinant *E. coli* expressed G (REG) proteins

Codon-optimized RSV G coding DNA for *E. coli* was chemically synthesized. *Not*I and *Pac*I restriction sites were used for cloning the RSV A2 G ectodomain coding sequence (amino acids 67 to 298) into the T7-based pSK expression vector for bacterial expression. DNA coding REG ΔCCD with residues 172-186 deleted and replaced with a (G4S)2 linker was prepared by a two-step overlapping PCR[7]. The deleted sequence contains the cysteine noose in addition to the CX3CR1 binding motif present in all RSV G proteins. The amplified DNA was digested with *Not*I and *Pac*I and ligated into the T7-based pSK expression vector for bacterial expression.

Recombinant RSV G 67-298 (REG 67-298) and REG ΔCCD proteins were expressed in *E. coli* BL21(DE3) cells (Novagen) and were purified as described previously[23,24]. Briefly, REG proteins expressed and localized in *E. coli* inclusion bodies (IB) were isolated by cell lysis, denatured and renatured by slowly diluting in redox folding buffer followed by dialysis. The dialysate was purified through a HisTrap FF

chromatography column (GE Healthcare). The protein concentrations were analyzed by bicinchoninic acid (BCA) assay (Pierce), and the purity of the recombinant G proteins from *E. coli* (REG) determined by SDS-PAGE. Linear peptides were synthesized chemically using Fmoc chemistry, purified by HPLC, conjugated to KLH, and dialyzed, as described before[6]

### Production of recombinant glycosylated G protein using 293 Flp-In cells (RMG)

The 293-Flp-In cell line (Cat. No. #R75007; ThermoFisher Scientific) stably expressing the G protein of either the RSV A2 G or the RSV B1 with secretory signal peptide from IgG kappa chain was developed as described previously. Briefly, 293-Flp-in cells were co-transfected with the plasmids expressing Flp-in recombinase and the RSV G ectodomain in DMEM media (Invitrogen). Twenty-four hours after transfection, culture medium was replaced with fresh DMEM containing 150 μg/mL of hygromycin for selection of stably transfected cells. For protein expression, cells were maintained in 293-Expression media (Invitrogen), and culture supernatant was collected every 3-4 days. The supernatant was cleared by centrifugation and filtered through a 0.45 μm filter before purification through a His-Trap FF column (GE healthcare).

### Monoclonal antibody production

To generate mouse hybridomas against RSV-G protein, two sets of 5 female C57BL/6 mice were immunized and boosted with recombinant non-glycosylated RSV G 67-298 from RSV-A2 strain (REG A2) or from RSV-B1 strains (REG B2) at 28 days apart. After the fusion of post-immunization splenocytes with mouse myeloma cell line, an initial screen of mouse hybridoma supernatants was performed against glycosylated RMG A2 and RMG B1 by ELISA. After the screening, the RSV G-binding positive hybridomas were single cell purified, and hybridomas screened again for clonality and screen for glycosylated RMG A2 or RMG B1 positive binders. All positive hybridoma clones were subjected for antibody production in serum-free media and MAbs were purified using protein A chromatography (GE Healthcare, Uppsala, Sweden).

### RSV G protein ELISA for MAb characterization

Immulon 2 HB 96-well microtiter plates were coated with 100 μl of purified recombinant G protein expressed in mammalian cells from either RSV-A2 (RMG-A2) or RSV-B1 (RMG-B1) in PBS (50 ng/well) per well at 4 °C overnight. After blocking with PBST containing 2% BSA, 100-fold dilutions of MAbs in blocking solution were added to each well, incubated for 1 h at RT, followed by addition of 5000-fold dilution of HRP-conjugated goat anti-mouse IgG-Fc specific antibody, and developed by 100 μl of OPD substrate solution. Absorbance was measured at 490 nm.

### Surface plasmon resonance

Steady-state equilibrium binding of MAbs was monitored at 25 °C using a ProteOn surface plasmon resonance (SPR) biosensor (Bio-Rad). The recombinant G protein from 293 T cells (REG-A2 delCCD) was coupled to a GLC sensor chip via amine coupling with 500 resonance units (RU) in the test flow channels. Biotinylated RSV-G peptides were captured using a NLC sensor chip. Samples of 100 μl of freshly prepared dilution of MAbs (1 μg/ml) were injected at a flow rate of 50 μl/min (contact duration, 120 seconds) for association. Disassociation was performed over a 600 s interval. Responses from the protein or peptide surface were corrected for the response from a mock surface and for responses from a buffer-only injection. Anti-CCR5 (2D7) MAb was used as a negative control. Total antibody binding and data analysis results were calculated with Bio-Rad ProteOn Manager software (version 3.0.1).

## Plaque reduction neutralization test

For the plaque reduction neutralization test (PRNT), heat-inactivated serum was diluted 4-fold and incubated with RSV-A2 virus (diluted to yield 20-50 plaques/well) containing 10% guinea pig complement (Rockland Immunochemical; Philadelphia, PA, USA) and incubated for 1 h at 37 °C. After incubation, 100 µl of the antibody-virus mixtures were inoculated in duplicate onto A549 monolayers in 48-well plates and incubated for 1 h at 37 °C. The inoculum was removed prior to adding the infection medium containing 0.8% methylcellulose. Plates were incubated for 5 to 7 days at which time the overlay medium was removed and cell monolayers fixed with 100% methanol; plaques were detected by immunostaining with rabbit RSV polyclonal anti-F sera, followed by addition of alkaline phosphatase goat anti-rabbit IgG (H + L) (Jackson, 111-055-144) antibody. The reactions were developed by using Vector Black Alkaline Phosphatase (AP) substrate kit (Vector Labs, Burlingame, CA). Numbers of plaques were counted per well and the neutralization titers were calculated by adding a trend line to the neutralization curves and using the following formula to calculate 50% endpoints: antilog of [(50+y-intercept)/slope].

## Ethics statement

All animal experiments were approved by the U.S. FDA Institutional Animal Care and Use Committee (IACUC) under Protocol #2009-20. The animal care and use protocol meets National Institutes of Health (NIH) guidelines.

## Mice RSV challenge study

Four- to 6-week-old female BALB/c mice (BALB/cAnNCr strain code #555) from Charles River Labs ($n = 5$ per group) were intraperitoneally (i.p.) injected with 20 µg/mouse of RSV G specific monoclonal antibodies, RSV G antibody (131-2 G), and phosphate-buffered saline (PBS, naive control). Mice were intranasally (i.n.) infected with $1 \times 10^6$ PFU/ ml of RSV A2 (rRSV-A2-L19-FFL) or RSV B1 (RSV-B1-FFL) under isoflurane anesthesia to determine the efficacy of protection and histopathological effects as previously described[24]. Mice were sacrificed by $CO_2$ asphyxiation 5 days post-RSV challenge (the day with peak viral load), and blood and lungs were collected. For determination of the viral load, the right lobe of the lung was collected.

## In vivo imaging of RSV-infected mice

Whole-body live imaging of infected mice was performed using IVIS imaging system as previously described[25]. In brief, mice were anesthetized in an oxygen-rich induction chamber with 2% isoflurane and administered 20 µl of RediJect D-Luciferin bioluminescent substrate (Perkin Elmer) intranasally. After a 5-min interval, mice were placed in the IVIS 200 Imaging systems (Xenocorp) equipped with the Living Image software (version 4.3.1.). Bioluminescence signals were recorded for 2 min for the whole body and for 1 min for lungs and nasal cavities, respectively. Images were analyzed with the LivingImage 4.5 software (PerkinElmer) according to manufacturer's instructions. Flux (photons) production using D-Luciferin bioluminescent substrate measured in IVIS requires expression of luciferase protein encoded by the genome of the recombinant RSV rA2-Line19F-Firefly Luciferase virus (rRSV-A2-L19-FFL) or RSV-B1-FFL virus expressing the firefly luciferase gene. Therefore, only infectious RSV particles that can infect cells in vivo are measured in the live imaging using IVIS. Inactivated virion particles that cannot infect cells are not measured in the flux live mice imaging assays.

## Lung viral titers by RSV immuno-plaque assay

Plaque formation units indicating RSV replication were visualized and quantified by immune-plaque assay with the Palivizumab antibody. Lung tissues were collected 5 days post challenge and individually measured lung RSV lung viral titer. Individual lungs (unperfused) were weighed and homogenized on ice in 1 mL DMEM, 2% FBS using an Omni tissue homogenizer. The clear supernatant was obtained by centrifugation at 3795×*g* for 10 min for a total of 2 centrifugations. Lung viral plaque-forming units (PFU) were determined by immune-plaque assay in Hep-2 cells. Media controls and lung homogenates mixed with RSV were incubated on HEp-2 cells for 1 h 37 °C, 5% CO2. After 4% formalin fixation, the plaques were detected with anti-F MAb (palivizumab from NIH Pharmacy; Catalog No- 1000509 at 1:1000 dilution) and then HRP conjugated anti-human IgG (Fc) antibodies (Jackson ImmunoResearch; Cat number 109-035-098 at 1:1000 dilution) were used. Stained and developed individual plaques were using DAB substrate kit (Invitrogen).

## Lung histopathology and inflammation scoring

The left lung was harvested from each individual mouse at 5 days post challenge and immediately fixed with 10% neutral buffered formalin. Lung samples were embedded in paraffin in the dorsoventral position. Subsequently, sections of tissue blocks were obtained and stained with hematoxylin and eosin (H & E) and analyzed under light microscopy[26]. For histopathological analysis the tissue slides were examined and scored blindly by a certified veterinary pathologist, including the following categories: epithelial alterations in alveolitis, bronchiolitis, perivascular, and interstitial space[26]. Inflammation and focal aggregates of infiltrating epithelial alveolar cells in the airways, blood vessel, and interstitial space were blindly examined, and measured using a semiquantitative scale (0 to 3) (0 = absent; normal), 1 (mild inflammation; <20% of lung affected), 2 (moderate inflammation; 20-40% of lung affected), and 3 = severe; 40-60% lung affected) by light microscope as previously described[6]. The scores were subsequently converted to a combined histopathology scale of 0-12.

## Statistical analysis

Statistical analyses were performed using GraphPad Prism version 8 (Graph Pad software Inc, San Diego, CA). Data were analyzed for significance using the student t-test, one-way ANOVA with Tukey's test for multiple comparisons, or a two-way ANOVA with Bonferroni posttests. The difference was considered statistically significant when the *P* value was <0.05. Correlations were calculated with a Spearman two-tailed test. *P* values <0.05 were considered significant with a 95% confidence interval.

## Reporting summary

Further information on research design is available in the Nature Portfolio Reporting Summary linked to this article.

# Data availability

All data are shown in the manuscript figures and supplementary information. The complete dataset for this study is provided in the Source Data file. There are restrictions to the availability of the MAbs described in this study due to US patent application. These Mabs are patented by the US FDA and are available under standard licensing agreement from FDA. All Mabs described in this study are covered under U.S. Patent Application No. 63/598,628 entitled: 'NEUTRALIZING AND PROTECTIVE MONOCLONAL ANTIBODIES AGAINST RESPIRATORY SYNCYTIAL VIRUS (RSV)'. This patent relates to Mabs and antigen-binding fragments that specifically bind to an attachment (G) protein of RSV and their use, for example, in methods of reducing RSV disease in a subject. All Mabs described in this study can be obtained under licensure from FDA. There are no other restrictions. Source data are provided with this paper.

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

## Acknowledgements

RSV-A2 line 19F-FFL expressing luciferase [rRSV-A2-L19-FFL] and RSV-B1 expressing firefly luciferase (RSV-B1-FFL) were provided by the. Dr. Martin L. Moore, (Emory University). We would like to thank Keith Peden and Basil Golding for a thorough review of the manuscript. The antibody characterization work described in this manuscript was supported by intramural FDA-CBER funds. The funders had no role in study design, data collection and analysis, interpretation, writing, decision to publish, or preparation of the manuscript. The content of this publication does not necessarily reflect the views or policies of the Department of Health and Human Services, nor does mention of trade names, commercial products, or organizations imply endorsement by the U.S. Government.

## Author contributions

All authors read and approved the final version of the manuscript. Designed research: S.K. Performed research: Y.L., L.K., E.C., G.G. and S.K. Contributed to writing: H.G. and S.K. S.K. and H.G. verified the underlying data.

## Competing interests

S.K. is the inventor of a patent application describing the monoclonal antibodies reported in this study. The rest of the authors declare that they have no competing interests.
