## [Peer Review File · Nature Communications]

Monoclonal antibodies targeting sites in respiratory syncytial virus attachment G protein provide protection against RSV-A and RSV-B in miceREVIEWER COMMENTS

Reviewer #1 (Remarks to the Author):

This is a well written manuscript reporting for the first time the protective effect of a panel of RSV-G monoclonal antibodies that target major antigenic sites on the G protein of RSV/A and RSV/B. The data were generated by internationally recognized scientist in RSV research, in particular, in the area of epitope mapping of the F and G proteins of RSV. In this report, monoclonal antibodies were generated in mice using a prime boost vaccination schedule of recombinant non-glycosylated G proteins produced in *E. coli* from either RSV/A2 or RSV/B1 strains. Mouse hybridomas were generated and screened for binding antibodies to G proteins, and further analyzed using epitope mapping by ELISA and SPR. The epitope mapping was based on G derived peptides from RSV/A2 that were previously identified from RSV genome fragment phage display library analysis of post-RSV infection sera of young children. Six classes of mAbs were generated: Class G0 (conformational-dependent), G1 (CCD: aa172-186), G2 (N-terminal domain: aa 61-90), G3 Strong stem domain around the CCD: aa129-152 & 169-207), G4 (weak stem domain around the CCD: aa129-152 & 169-207), and G5 (C-terminal domain: 169-297). Some of the mAbs in class G0, G1, G3, and G5 had significant cross-reactivity to both RSV/A and RSV/B glycosylated form of G protein. As expected, none of the panel of mAbs had neutralizing antibody activity in vitro. The efficacy of the mAbs was determined in a prophylactic mouse model against either an RSV/A or RSV/B challenge. Balb-c mice were administered 20 µg of a G mAb and 24 hours later were intranasally infected with 10⁶ RSV-A2-L19-FFL or RSV-B1 FFL. Endpoints were daily assessment of fluorescence using whole body live imaging until euthanasia (Day-5), RSV viral titer of lung and nose at Day 5, and lung histopathology at Day 5. A positive control mAb, 131-2G that targets the CCD and with established efficacy, was used for comparison to the other G mAbs. Overall, 18 mAbs were generated, of which 5 and 6 of the mAbs belonged to class G0 and G1, respectively. The other classes were composed of either 1 or 2 mAbs. None of the mAbs reduced infection in the nose either measured by a plaque assay or nasal flux activity. All mAbs had excellent reduction in lung viral titers, however, when measured by lung flux activity only some of the mAbs in G0, G1 and G5 had significant reduction. The same mAbs also had reduction in lung pathology. Interestingly a significant correlation was observed between flux activity of the lung and pulmonary histopathology, but not infectious titer measured by a plaque assay and pulmonary histopathology. The manuscript concludes that cross-reactive G mAbs may merit further development for the prevention of RSV infection.

General Comments.

This is a well written manuscript reporting on novel data around a new class of G mAbs with cross-reactivity against RSV/A and RSV/B infection in a mouse model. The data is scientifically sound and expands on our knowledge of the G protein, its dominant antigenic sites and new classes of protective mAbs. Equally important there was no evidence of antibody enhanced pathology or infection. Although the mouse model is an excellent screening model to study efficacy and pathology of new drugs against RSV, it is semi-permissive to RSV and normally requires a large inoculum to induce infection. The authors are correct that their G mAbs merit further development, and the cotton rat should be considered in the next stage of development because of its enhance permissiveness to RSV infection compared to the mouse model. Also, contemporary RSV strains from the RSV/A/Ontario and RSV/B/Buenos Aires genotypes should be considered compared to the prototypic viruses that were isolated in the 1960s. Other pre-clinical models such as the human airway organoid (lung or nose) that may better recapitulate the human airway should be considered in the evaluation of the G mAbs.

Specific comments.

Abstract. Nirsevimab is also license for healthy infants under 8 months of age. Please revise statement that states only high-risk infants.

Introduction. Nirsevimab is also licensed by the FDA and approved by ACIP for use in healthy infants 8 months of age and younger. Please revise pg 3, lines 55-56.

Please provide a supplementary table that shows the mAbs were generated by the prime boost vaccination schedule using recombinant non-glycosylated G proteins of RSV/A2 and RSV/B1. It is interesting to know if they generated similar or different classes of G mAbs. Alternatively, this information could be included in Table 1.

Pg 6 lines 114-116. Lease revise the statement since class G2 mAbs did bind to the CCD-deleted REG-A2 protein in SPR (see Table 1).

Pg 8, line 171: should be 22B11 and not 2B11.

Pg 9, lines 181-183, please consider revising because various classes of mAbs did not reduce lung pathology even though they targets comparable antigenic sites to those mAbs that did reduce lung pathology.

Pg 9, line 198: should be 36E10 and not 3E10.

Pg 10, lines 205-210. Should also present on what appears to be a lack of correlation between lung PFU and lung flux activity. The authors should mention this in the discussion section indicating that infectious RSV as measured by PFU may not be a reliable measure of infectivity as compared to flux activity in the mouse model.

Histopathology analysis: It would be informative if additional information on the parameters that drove the histopathology score were provided. Was the pathology score driven by one or two parameters? Please add information on the composition of the score by mAb, perhaps that could be included in a supplementary table or figure.

In the discussion section, please discuss why there was such a large difference in lung pathology when RSV/A2 compared to RSV/B1 was used as the infecting virus. That is very interesting and I am not aware that such an observation has been previously reported.

Figure 5 consider revising. Perhaps illustrating ranking separately by efficacy based on lung flux activity and lung pathology.

Reviewer #2 (Remarks to the Author):

NCOMMS-23-50260-

mAbs were generated from mice immunized with recombinant non-glycosylated G proteins from either RSV A2 strain or B1. mAbs binding sites identified 5 antigenic classes (G0-G5). None of the anti-G mAbs neutralized RSV A2 or B1 in vitro. The protective efficacy of the MAb was determined in mice challenged with either A2 line 19F-luciferase or RSV-B1-luciferase one day after anti-G mAb treatment, and virus titers, dissemination, and pathology were determined in lungs on day 5 post-challenge. All mAbs reduced viral titers, and most mAbs reduced lung pathology compared to mock-treated animals. RSV dissemination by whole-body imaging was variable but correlated with lung pathology following the challenge.

This study was very well performed and supports earlier studies on anti-G mAbs. The study reported here does offer new insights regarding novel anti-G mAbs targeting multiple sites on G protein and defines mAbs that provide cross-reactive protection following challenges with either RSV A or B.

Better clarification is needed regarding virus transmission studies in the nasal cavity as measured by Flux units using live imaging, such as the factors known to affect flux, whether

an inactivated virus is also measured, etc. Also, please clarify the correlation between lung pathology and RSV dissemination in the lungs, specifically clarify the potential mechanisms of virus dissemination to the lungs. Briefly speculate if the same mechanisms may occur for anti-F mAb treatment and if there is a role for combining anti-G/F mAbs in treatment.

Reviewer #3 (Remarks to the Author):

Monoclonal antibodies targeting sites in RSV attachment G protein provide protection against RSV A and RSV B

Lee et al.

This is a study to characterize a set of newly developed monoclonal antibodies against the G attachment protein of RSV. The antibody set is characterized by ELISA, epitope specific binding by surface plasmon resonance, neutralization assay, and by prophylactic treatment to determine protection against RSV and B challenge in the mouse model. In vivo, efficacy is quantified in the lung by plaque assay, lung histopathology and lung viral dissemination using a previous developed in vivo method based in the detection of luciferase virus.

The manuscript is well presented, written, with figures mostly clear, and results in many cases supporting conclusions and claims. The generation of new sets of monoclonal antibodies is important and the methodology for mapping their target epitope is the best in the field. These new antibodies against the RSV G protein, as is the case for 131-2G, does not generate measurable neutralizing antibodies, however, they clearly show inhibition of viral replication in vivo.

The manuscript however felt short to convincingly demonstrate superiority of any of the new antibodies over 131-2G. There is stronger reduction in lung pathology by many of them but not significant when compared to 131-2G. In addition, the preclinical studies presented are preliminary in nature.

Figure 4 shows significant correlations between pathology and viral dissemination, and not significant correlation between pathology and viral titers in the lung. This data point to one main claim in the work: that the parameter of virus dissemination in the lung (measured by in vivo imaging in Flux units) directly correlates with pathology. I would argue that this the data presented is weak to support this assumption and that these correlations might not be completely valid due to the potential broad range of action of the different antibodies tested (as discussed in the text). In fact, for some antibodies, it seems that the correlation is inverse (e.g., 12G11 shows low flux but mean pathology seems higher for RSV A). Would it be of more value to draw correlations for each antibody treatment and pathology for each animal (probably need to increase n) or with lung cytokine production?

Minor Points:

Line 115: According to Table 1, 7C6 and 7G6 bind REG-A2 delICCD in contrast to the statement.

Line 167: It would be suggested to show how the different antibodies modulate different lung pathology categories (e.g., bronchiolitis, alveolitis, etc.).

178: Add "pathology scores" - between "Lung.....of mice treated"

Line 207: The description points to the wrong figures.

Would be important to show visual examples of differences in bioluminescence from different groups of mice.

Figure 4: It is difficult to identify each antibody symbols in the correlation graphs.

REVIEWER COMMENTS:

Reviewer #1:

This is a well written manuscript reporting for the first time the protective effect of a panel of RSV-G monoclonal antibodies that target major antigenic sites on the G protein of RSV/A and RSV/B. The data were generated by internationally recognized scientist in RSV research, in particular, in the area of epitope mapping of the F and G proteins of RSV. In this report, monoclonal antibodies were generated in mice using a prime boost vaccination schedule of recombinant non-glycosylated G proteins produced in *E. coli* from either RSV/A2 or RSV/B1 strains. Mouse hybridomas were generated and screened for binding antibodies to G proteins, and further analyzed using epitope mapping by ELISA and SPR. The epitope mapping was based on G derived peptides from RSV/A2 that were previously identified from RSV genome fragment phage display library analysis of post-RSV infection sera of young children. Six classes of mAbs were generated: Class G0 (conformational-dependent), G1 (CCD: aa172-186), G2 (N-terminal domain: aa 61-90), G3 Strong stem domain around the CCD: aa129-152 & 169-207), G4 (weak stem domain around the CCD: aa129-152 & 169-207), and G5 (C-terminal domain: 169-297). Some of the mAbs in class G0, G1, G3, and G5 had significant cross-reactivity to both RSV/A and RSV/B glycosylated form of G protein. As expected, none of the panel of mAbs had neutralizing antibody activity in vitro. The efficacy of the mAbs was determined in a prophylactic mouse model against either an RSV/A or RSV/B challenge. Balb-c mice were administered 20 µg of a G mAb and 24 hours later were intranasally infected with 10⁶ RSV-A2-L19-FFL or RSV-B1 FFL. Endpoints were daily assessment of fluorescence using whole body live imaging until euthanasia (Day-5), RSV viral titer of lung and nose at Day 5, and lung histopathology at Day 5. A positive control mAb, 131-2G that targets the CCD and with established efficacy, was used for comparison to the other G mAbs. Overall, 18 mAbs were generated, of which 5 and 6 of the mAbs belonged to class G0 and G1, respectively. The other classes were composed of either 1 or 2 mAbs. None of the mAbs reduced infection in the nose either measured by a plaque assay or nasal flux activity. All mAbs had excellent reduction in lung viral titers, however, when measured by lung flux activity only some of the mAbs in G0, G1 and G5 had significant reduction. The same mAbs also had reduction in lung pathology. Interestingly a significant correlation was observed between flux activity of the lung and pulmonary histopathology, but not infectious titer measured by a plaque assay and pulmonary histopathology. The manuscript concludes that cross-reactive G mAbs may merit further development for the prevention of RSV infection.

Response: We thank the reviewer for the positive response and appreciating our study.

General Comments.

This is a well written manuscript reporting on novel data around a new class of G mAbs with cross-reactivity against RSV/A and RSV/B infection in a mouse model. The data is scientifically sound and expands on our knowledge of the G protein, its dominant antigenic sites and new classes of protective mAbs. Equally important there was no

evidence of antibody enhanced pathology or infection. Although the mouse model is an excellent screening model to study efficacy and pathology of new drugs against RSV, it is semi-permissive to RSV and normally requires a large inoculum to induce infection. The authors are correct that their G mAbs merit further development, and the cotton rat should be considered in the next stage of development because of its enhance permissiveness to RSV infection compared to the mouse model. Also, contemporary RSV strains from the RSV/A/Ontario and RSV/B/Buenos Aires genotypes should be considered compared to the prototypic viruses that were isolated in the 1960s. Other pre-clinical models such as the human airway organoid (lung or nose) that may better recapitulate the human airway should be considered in the evaluation of the G mAbs.

Response: We agree with reviewer thoughts and suggestion. We have added the following to discussion section (lines 267-272 in tracked manuscript):

'In the next stage of development, the findings in the mice challenge studies will be expanded to the cotton rat model with contemporary RSV strains from the RSV/A/Ontario and RSV/B/Buenos Aires genotypes to understand the prophylactic and therapeutic application of these MAbs. The protective mechanism of these anti-G MabsMAbs can be further deciphered in other pre-clinical models such as the human airway organoid (lung or nose) that may better recapitulate the human airway.'

Specific comments.

Abstract. Nirsevimab is also license for healthy infants under 8 months of age. Please revise statement that states only high-risk infants.

Response: We removed 'high risk' from abstract.

Introduction. Nirsevimab is also licensed by the FDA and approved by ACIP for use in healthy infants 8 months of age and younger. Please revise pg 3, lines 55-56.

Response: The statement has been revised to:

'There are two licensed monoclonal antibodies (MAbs), Palivizumab and Nirsevimab, both target the fusion (F) protein, which can reduce disease in high-risk premature-birth infants or healthy infants 8 months of age and younger, respectively, if administered prior to RSV infection (ref 7).'

Please provide a supplementary table that shows the mAbs were generated by the prime boost vaccination schedule using recombinant non-glycosylated G proteins of RSV/A2 and RSV/B1. It is interesting to know if they generated similar or different classes of G mAbs. Alternatively, this information could be included in Table 1.

Response: The information on immunogen has been added to supplementary table S1.

Pg 6 lines 114-116. Lease revise the statement since class G2 mAbs did bind to the CCD-deleted REG-A2 protein in SPR (see Table 1).

Response: The statement has been revised to:

'Class G2 included two MAbs (7C6 and 7G6) that reacted with RMG-A2, but not to RMG-B1 in ELISA. These class G2 MAbs reacted strongly to the CCD-deleted REG-A2 protein in SPR and bind primarily to N-terminal peptide (aa residues 61-90) of RSV-G.'

Pg 8, line 171: should be 22B11 and not 2B11.

Response: Thanks for catching the typo. It has been corrected to 22B11.

Pg 9, lines 181-183, please consider revising because various classes of mAbs did not reduce lung pathology even though they targets comparable antigenic sites to those mAbs that did reduce lung pathology.

Response: The statement has been revised to (lines 180-186 in tracked manuscript):

Lung pathology scores of mice treated with few MAbs including G1-MAb 40D8 were similar to those observed for uninfected control animals following RSV-B1 challenge.

Together, these finding suggested that the majority of Mabs belonging to various classes, targeting different regions of RSV-A2 and RSV-B1 G proteins given one day prior to challenge, reduced lung pathology to different levels following either RSV-A2 or RSV-B1 challenge in the mouse model.

Pg 9, line 198: should be 36E10 and not 3E10.

Response: Thanks for catching the typo. It has been corrected to 36E10.

Pg 10, lines 205-210. Should also present on what appears to be a lack of correlation between lung PFU and lung flux activity. The authors should mention this in the discussion section indicating that infectious RSV as measured by PFU may not be a reliable measure of infectivity as compared to flux activity in the mouse model.

Response: The correlation analysis between lung pfu and lung flux activity has been added in new supplementary figure S4. The results have been added in lines 213-214 in tracked manuscript:

'No correlation was observed between lung PFU and lung flux activity at day 5 post-RSV challenge in these mice infected with either RSV-A2 or RSV-B1 (Figure S4).'

Histopathology analysis: It would be informative if additional information on the parameters that drove the histopathology score were provided. Was the pathology score driven by one or two parameters? Please add information on the composition of the score by mAb, perhaps that could be included in a supplementary table or figure.

Response: The data on individual histopathology scores has been added in new supplementary figure S2. The results have been added in lines 166-169 in the tracked manuscript:

'Inflammation and focal aggregates of infiltrating cells were examined and measured using a semiquantitative scale (0 to 3) (0 = absent; normal), 1 (mild inflammation; <20% of lung affected), 2 (moderate inflammation; 20-40% of lung affected), and 3 = severe; 40-60% lung affected) by light microscope (Figure S2).'

In the discussion section, please discuss why there was such a large difference in lung pathology when RSV/A2 compared to RSV/B1 was used as the infecting virus. That is very interesting and I am not aware that such an observation has been previously reported.

Response: The combined lung pathology scores (scale 0-12) between animals infected with either RSV/A2 or RSV/B1 are between 7-10 as shown in Figure 2c and 2d (PBS group) and are not significantly different.

Figure 5 consider revising. Perhaps illustrating ranking separately by efficacy based on lung flux activity and lung pathology.

Response: We have shown individual data for lung flux and lung pathology in Figures 2 and 3. To succinctly summarize the totality of different feature outcomes for each MAb and provide the ranking for different Mabs targeting various antigenic sites in G, we believe the figure 5 shows the relevant data appropriately. As per reviewer's suggestion, we have modified the figure legend to clarify it further.

Figure 5: Cross-reactivity and protective efficacy of different classes of anti-G MAbs against RSV-A and RSV-B. Schematic summarizing the classification of RSV-G binding MAbs by mapping studies and protective efficacy based on lung flux and lung pathology in BALB/c mice challenge studies against RSV-A2 and RSV-B1. All MAbs reduced lung infectious viral titers. High protective MAbs were defined as the MAbs that significantly reduced lung fluxes and lung pathology; moderately protective MAbs were defined as those who reduced lung flux and/or lung pathology (similar to previously described moderately protective MAb 131-3G), but less efficiently than the highly protective Mabs category; while low protective MAbs only reduced lung infectious viral titers but did not reduce lung flux nor lung pathology.

Reviewer #2:

mAbs were generated from mice immunized with recombinant non-glycosylated G proteins from either RSV A2 strain or B1. mAbs binding sites identified 5 antigenic classes (G0-G5). None of the anti-G mAbs neutralized RSV A2 or B1 in vitro. The protective efficacy of the MAbs was determined in mice challenged with either A2 line 19F-luciferase

or RSV-B1- luciferase one day after anti-G mAb treatment, and virus titers, dissemination, and pathology were determined in lungs on day 5 post-challenge. All mAbs reduced viral titers, and most mAbs reduced lung pathology compared to mock-treated animals. RSV dissemination by whole-body imaging was variable but correlated with lung pathology following the challenge.

This study was very well performed and supports earlier studies on anti-G mAbs. The study reported here does offer new insights regarding novel anti-G mAbs targeting multiple sites on G protein and defines mAbs that provide cross-reactive protection following challenges with either RSV A or B.

Response: We thank the reviewer for appreciating our study and encouraging comments to make our manuscript better.

Better clarification is needed regarding virus transmission studies in the nasal cavity as measured by Flux units using live imaging, such as the factors known to affect flux, whether an inactivated virus is also measured, etc.

Response: We have earlier published (ref. 22) on the live imaging of mice model following RSV infection and its use for evaluation of vaccines and therapeutics. This publication is cross-referenced at several places throughout the manuscript (ref. 22). In this publication, we showed correlation between flux, virus infectious titers and RSV RNA by qRT-PCR. The inactivated virus does not produce flux signal in animals.

We have clarified it in lines 400-405 in the tracked revised manuscript:

'Flux (photons) production using D-Luciferin bioluminescent substrate measured in IVIS requires expression of luciferase protein encoded by the genome of the recombinant RSV rA2-Line19F-Firefly Luciferase virus (rRSV-A2-L19-FFL) or RSV-B1-FFL virus expressing the firefly luciferase gene. Therefore, only infectious RSV particles that can infect cells in vivo are measured in the live imaging using IVIS. Inactivated virion particles that cannot infect cells are not measured in the flux live mice imaging assays.'

Also, please clarify the correlation between lung pathology and RSV dissemination in the lungs, specifically clarify the potential mechanisms of virus dissemination to the lungs.

Response: The correlation and potential mechanisms have been mentioned and discussed.

lines 210-214 in the tracked manuscript:

'A statistically significant correlation was observed between the lung pathology scores and lung flux measurement on day 5 for individual mice across all groups for RSV-A2 ($p=0.0317$) (Fig. 4a) and RSV-B1 ($p= 0.0058$) (Fig. 4b), but not with lung infectious viral titers (Figure 4c and d). No correlation was observed between lung PFU and lung flux activity at day 5 post-RSV challenge in these mice infected with either RSV-A2 or RSV-B1 (Figure S4).'

Lines 250-253 in the tracked manuscript:

The correlation between lung pathology and RSV dissemination (lung flux) but not with infectious viral titer in the lungs is remarkable (Figure 4) and suggest that viral dissemination may be a more sensitive readout in the mouse model.

Lines 258-267 in the tracked manuscript:

Our observations suggests that the lung pathology following RSV infection could be a result of viral dissemination primarily by cell-to-cell spread in the lungs and the strong inflammatory response that follows. RSV spread within the host more efficiently via direct cell-to-cell transmission that may result in the lung pathology observed in animals in the absence of the productive infectious viral particle generated following RSV infection (ref 24). Several of the new anti-G MAbs were more efficient in restricting the viral dissemination in lungs as measured by live imaging of RSV challenged mice, suggesting a key role for G protein in the cell-to-cell spread of RSV.

Briefly speculate if the same mechanisms may occur for anti-F mAb treatment and if there is a role for combining anti-G/F mAbs in treatment.

Response: We have discussed potential mechanisms and potential anti-F combination in lines 275-281 in the tracked revised manuscript:

'The mechanisms of protection from viral dissemination to the lungs by the new anti-G MAbs, similar to MAb 131-2G, are not well understood, but may involve Fc mediated functions including antibody-dependent cellular cytotoxicity (ADCC), antibody-dependent cellular phagocytosis (ADCP), and antibody-dependent complement deposition (ADCD), similar to some anti-F antibodies elicited by infection or vaccination (ref 29). Similar mechanisms may occur for anti-F MAb treatment and therefore, our study suggests a potential role for combining anti-G and anti-F MAbs as more effective prophylaxis and/or treatment against RSV.'

Reviewer #3:

Monoclonal antibodies targeting sites in RSV attachment G protein provide protection against RSV A and RSV B

Lee et al.

This is a study to characterize a set of newly developed monoclonal antibodies against the G attachment protein of RSV. The antibody set is characterized by ELISA, epitope specific binding by surface plasmon resonance, neutralization assay, and by prophylactic treatment to determine protection against RSV and B challenge in the mouse model. In vivo, efficacy is quantified in the lung by plaque assay, lung histopathology and lung viral

dissemination using a previously developed in vivo method based on the detection of luciferase virus.

The manuscript is well presented, written, with figures mostly clear, and results in many cases supporting conclusions and claims. The generation of new sets of monoclonal antibodies is important and the methodology for mapping their target epitope is the best in the field. These new antibodies against the RSV G protein, as is the case for 131-2G, does not generate measurable neutralizing antibodies, however, they clearly show inhibition of viral replication in vivo.

The manuscript however felt short to convincingly demonstrate superiority of any of the new antibodies over 131-2G. There is stronger reduction in lung pathology by many of them but not significant when compared to 131-2G. In addition, the preclinical studies presented are preliminary in nature.

Figure 4 shows significant correlations between pathology and viral dissemination, and not significant correlation between pathology and viral titers in the lung. This data point to one main claim in the work: that the parameter of virus dissemination in the lung (measured by in vivo imaging in Flux units) directly correlates with pathology. I would argue that this data presented is weak to support this assumption and that these correlations might not be completely valid due to the potential broad range of action of the different antibodies tested (as discussed in the text). In fact, for some antibodies, it seems that the correlation is inverse (e.g., 12G11 shows low flux but mean pathology seems higher for RSV A). Would it be of more value to draw correlations for each antibody treatment and pathology for each animal (probably need to increase n) or with lung cytokine production?

Response: As mentioned by the reviewer, some Mabs trended to better than 131-2G in lung pathology (77D2, 40D8, etc) and lung fluxes (77D2, 40D8, 12G11, 7H11 etc), however, these differences did not reach statistical significance. Moreover, combining these multiple parameters including lung pathology and lung fluxes, as summarized in figure 5, 131-2G was categorized as moderately protective, while some Mabs targeting different antigenic sites in RSV-G was found to be superior to 131-2G, and were in the highly protective (77D2, 40D8, 7H11 etc) category. To make this point clearly, we have revised the statements and figure 5 legend.

Lines 174-182 in the tracked revised manuscript:

Interestingly, class G0 MAb 77D2, G1 MAb 40D8, 1D9 and 36E10 as well as G2 MAb 7G6 treated animals showed lower pathology than 131-2G treated animals following RSV-A2 challenge, although these differences did not reach statistical significance (Figure 2c).

All mice that received prophylactic treatment of MAb prior to RSV-B1 challenge demonstrated reduced lung pathology scores, similar to MAb 131-2G treated animals compared with lung pathology observed in the PBS control treated animals (Figure 2d). Lung pathology scores of mice treated with few MAb

including G1-MAb 40D8 showed stronger reduction and were similar to those observed for uninfected control animals following RSV-B1 challenge.

Lines 199-202 in the tracked revised manuscript:

Importantly, in RSV-A2 challenged animals, several MAbs reduced day 5 lung-fluxes more efficiently than 131-2G, including class G0 MAb 77D2, G1 MAbs 40D8, 1D9, 12G11, 22B11 and 36E10, G3 MAb 48E2, as well as G5 MAb 7H11 in RSV-A2 infected animals (Figure 3c).

Lines 215-219 in the tracked revised manuscript:

Together, our data demonstrates that several anti-G MAbs targeting multiple sites, including conformation-dependent class G0 MAb 77D2, CCD-specific class G1 MAb 40D8, and carboxy terminus of CCD class G5 MAb 7H11, showed cross-reactive protection from lung pathology and RSV dissemination following challenge with either RSV-A or B subtypes better than MAb 131-2G (Figure 5).

Figure 5: Cross-reactivity and protective efficacy of different classes of anti-G MAbs against RSV-A and RSV-B. Schematic summarizing the classification of RSV-G binding MAbs by mapping studies and protective efficacy based on lung flux and lung pathology in BALB/c mice challenge studies against RSV-A2 and RSV-B1. All MAbs reduced lung infectious viral titers. High protective MAbs were defined as the MAbs that significantly reduced lung fluxes and lung pathology; moderately protective MAbs were defined as those who reduced lung flux and/or lung pathology (similar to previously described moderately protective MAb 131-3G), but less efficiently than the highly protective MAbs category; while low protective MAbs only reduced lung infectious viral titers but did not reduce lung flux nor lung pathology.

For the correlations, we have shown data for each antibody treatment and lung pathology vs lung flux (figure 4a-b), lung pathology vs lung viral load (figure 4c-d) and lung viral load vs lung flux (new Figure S4) for each animal. These correlations provide the importance of measuring different parameters following RSV infection to evaluate the MAb therapeutics. We think that this is more appropriate correlation to demonstrate for all MAbs together.

Lines 210-214 in the tracked revised manuscript:

A statistically significant correlation was observed between the lung pathology scores and lung flux measurement on day 5 for individual mice across all groups for RSV-A2 ($p=0.0317$) (Fig. 4a) and RSV-B1 ($p= 0.0058$) (Fig. 4b), but not with lung infectious viral titers (Figure 4c and d). No correlation was observed between lung PFU and lung flux activity at day 5 post-RSV challenge in these mice infected with either RSV-A2 or RSV-B1 (Figure S4).

Minor Points:

Line 115: According to Table 1, 7C6 and 7G6 bind REG-A2 delCCD in contrast to the statement.

Response: The statement has been revised to:

‘Class G2 included two MAbs (7C6 and 7G6) that reacted with RMG-A2, but not to RMG-B1 in ELISA. These class G2 MAbs reacted strongly to the CCD-deleted REG-A2 protein in SPR and bind primarily to N-terminal peptide (aa residues 61-90) of RSV-G.’

Line 167: It would be suggested to show how the different antibodies modulate different lung pathology categories (e.g., bronchiolitis, alveolitis, etc.).

Response: As suggested by the reviewer, data on individual histopathology scores has been added in new supplementary figure S2. The results have been added in lines 166-169 in the tracked manuscript:

‘Inflammation and focal aggregates of infiltrating cells were examined and measured using a semiquantitative scale (0 to 3) (0 = absent; normal), 1 (mild inflammation; <20% of lung affected), 2 (moderate inflammation; 20-40% of lung affected), and 3 = severe; 40-60% lung affected) by light microscope (Figure S2).’

178: Add “pathology scores” - between “Lung.....of mice treated”

Response: Statement has been revised in lines 166-169 in the tracked manuscript: Lung pathology scores of mice treated with few MAbs including G1-MAb 40D8 showed stronger reduction and were similar to those observed for uninfected control animals following RSV-B1 challenge.

Line 207: The description points to the wrong figures.

Response: We thank the reviewer; the panels have been switched in figure 4 to match the text.

Would be important to show visual examples of differences in bioluminescence from different groups of mice.

Response: As suggested by the reviewer, representative *in vivo* imaging of RSV infection in mice has been added in new supplementary figure S3 and referred in lines 195-196 in the tracked manuscript:

‘On day 5, the lung fluxes varied among the MAb- treated animals (Figure S3).’

Figure 4: It is difficult to identify each antibody symbols in the correlation graphs.

Response: The correlation figures in figure 4 depict the same color symbols for each MAb in all the main figures throughout the manuscript to keep them consistent for easy cross-reference, with appropriate symbol labels. The figure

panel is provided in the maximum possible size to fit dimensions for one figure as per manuscript guidelines.

REVIEWERS' COMMENTS

Reviewer #1 (Remarks to the Author):

The authors have responded to the comments raised by the reviewers appropriately. I have no further comment.

Reviewer #2 (Remarks to the Author):

The authors satisfactorily addressed the reviewer's concerns. The work was solid and significant, and the methodology was excellent.

Reviewer #3 (Remarks to the Author):

The manuscript has been corrected and it is acceptable for publication.

REVIEWER's COMMENTS:

Reviewer #1 (Remarks to the Author):

The authors have responded to the comments raised by the reviewers appropriately. I have no further comment.

Response: We thank the reviewer for the positive response to our revised manuscript.

Reviewer #2 (Remarks to the Author):

The authors satisfactorily addressed the reviewer's concerns. The work was solid and significant, and the methodology was excellent.

Response: We thank the reviewer for appreciating our study.

Reviewer #3 (Remarks to the Author):

The manuscript has been corrected and it is acceptable for publication.

Response: We thank the reviewer for accepting our revised manuscript.